# Advancing Personalized Medicine by Analytical Means: Selection of Three Metabolites That Allows Discrimination between Glaucoma, Diabetes, and Controls

**DOI:** 10.3390/metabo14030149

**Published:** 2024-02-29

**Authors:** David Bernal-Casas, Joan Serrano-Marín, Juan Sánchez-Navés, Josep M. Oller, Rafael Franco

**Affiliations:** 1Department of Genetics, Microbiology and Statistics, Faculty of Biology, University of Barcelona, 08028 Barcelona, Spain; bernalcd@ub.edu (D.B.-C.); joller@ub.edu (J.M.O.); 2Department of Biochemistry and Molecular Biomedicine, Faculty of Biology, University of Barcelona, 08028 Barcelona, Spain; joan.serrano.marin@ub.edu; 3Department of Ophthalmology, Ophthalmedic and I.P.O, Institute of Ophthalmology, 07011 Palma de Mallorca, Spain; juansanchez.naves@gmail.com; 4CiberNed, Network Centre for Neurodegenerative Diseases, National Spanish Health Institute Carlos III, 28029 Madrid, Spain; 5School of Chemistry, University of Barcelona, 08028 Barcelona, Spain

**Keywords:** aqueous humour, diagnostic tools, Parkinson’s disease, metabolomic signatures, precision medicine, advanced statistical, methods, bioanalysis, discrimination tools

## Abstract

This paper aimed at devising an intelligence-based method to select compounds that can distinguish between open-angle glaucoma patients, type 2 diabetes patients, and healthy controls. Taking the concentration of 188 compounds measured in the aqueous humour (AH) of patients and controls, linear discriminant analysis (LDA) was used to identify the right combination of compounds that could lead to accurate diagnosis. All possibilities, using the leave-one-out approach, were considered through ad hoc programming and in silico massive data production and statistical analysis. Our proof of concept led to the selection of four molecules: acetyl-ornithine (Ac-Orn), C3 acyl-carnitine (C3), diacyl C42:6 phosphatidylcholine (PC aa C42:6), and C3-DC (C4-OH) acyl-carnitine (C3-DC (C4-OH)) that, taken in combination, would lead to a 95% discriminative success. 100% success was obtained with a non-linear combination of the concentration of three of these four compounds. By discarding younger controls to adjust by age, results were similar although one control was misclassified as a diabetes patient. Methods based on the consideration of individual clinical chemical parameters have limitations in the ability to make a reliable diagnosis, stratify patients, and assess disease progression. Leveraging human AH metabolomic data, we developed a procedure that selects a minimal number of metabolites (3–5) and designs algorithms that maximize the overall accuracy evaluating both positive predictive (PPV) and negative predictive (NPV) values. Our approach of simultaneously considering the levels of a few metabolites can be extended to any other body fluid and has potential to advance precision medicine. Artificial intelligence is expected to use algorithms that use the concentration of three to five molecules to correctly diagnose diseases, also allowing stratification of patients and evaluation of disease progression. In addition, this significant advance shifts focus from a single-molecule biomarker approach to that of an appropriate combination of metabolites.

## 1. Introduction

Biomarker discovery is a hot topic, both for acute and chronic diseases. Neurodegeneration is probably one of the most orphaned in terms of suitable biomarkers for reliable patient stratification, assessment of disease progression, and evaluation of the efficacy of new therapies [1,2]. Also relevant is the possibility of discovering biomarkers of comorbidities.

Proteomics and metabolomics are the two most common methods for routinely identifying molecules that could serve as biomarkers. One of the most recent developments in metabolomics combines novel techniques for the derivatization of small compounds with identification and quantification by mass spectrometry [3,4]. Hence, it is possible to measure the concentration of hundreds of compounds using a few microliters of body fluids regardless of their complexity, that is, being useful for serum, urine, sweat, cerebrospinal fluid, saliva, urine, and aqueous humour.

In particular, the range of concentrations in metabolomics approaches using human body fluids is very wide; that is, it is possible to reliably determine the concentration of minor compounds, such as acyl-carnitines, which can be found at nM levels, as well as those that are abundant, such as amino acids, which can reach mM levels. On the one hand, metabolomics approaches can establish reference values for up to 500 parameters (concentration of small compounds, MW < 1000 kDa) in serum. On the other hand, data analysis is a major issue due to both the generation of large numbers of parameters, each corresponding to a sample and a concentration of molecules, and the intrinsic variability found in human samples [3,4].

Classically, clinical chemistry approaches were based on comparing one molecule with one disease, for example, plasma glucose concentration with diabetes. In the case of transcriptomic-based studies, data normalization is required. In contrast, in metabolomics, there is no need to normalize metabolite concentrations even when hundreds of compounds are determined in two conditions, for example, healthy control versus type 2 diabetes patients or type 2 diabetes patients versus glaucoma patients.

In fact, we have recently identified new biomarkers after comparing the concentration of small metabolites in the aqueous humour (AH) of healthy controls, patients with open-angle glaucoma, and patients with type 2 diabetes. Instead of glucose, one molecule, acetyl-ornithine, makes it possible to detect if a given AH is from a patient with type 2 diabetes or a healthy control [5].

Artificial intelligence must be loaded with data and strategies that allow healthcare providers to offer more personalized, precise, and effective treatments. Clinical chemistry has been an ally of doctors for decades. The first advances were due to the determination of small molecules in blood/plasma and urine and the establishment of biomarkers that are still used today; the main example is the glucose level in plasma for the diagnosis and management of diabetes. There is a need of novel biomarkers for, among others, neurodegenerative diseases; in the last two decades translational research has focused more on proteomics and transcriptomics than on metabolomics.

While metabolomics tools have demonstrated high sensitivity in detecting critical metabolites involved in pathological conditions, the trends of many of these relevant metabolites are similar among diseases, making the interpretation challenging. Computational tools developed and improved under the umbrella of advanced statistical methods, machine learning, and/or artificial intelligence are needed to find the minimum number of metabolites that allow distinguishing between different pathologies, even those with similar metabolic patterns [6,7,8,9]. However, the challenge is to design an intelligent technique that unbiasedly converts a combination of parameters into a reliable diagnosis.

Importantly, advanced mathematical methods ranging from simple regression analyses to complex deep learning architectures are becoming prevalent in healthcare. Apart from early illness detection, intelligent techniques are explored in drug discovery and patient risk assessment. Medical data from, among others, ultrasound, MRI, mammography, genomics, or positron emission tomography would feed any approach aimed at benefiting patient diagnosis and treatment [10,11,12,13].

The objective of this study was to verify whether it is possible to implement an intelligent-based tool to select a limited number of compounds so that, together, their concentration in a specific body fluid allows the unequivocal diagnosis of a particular disease. The strategy has two components, one is the determination of compounds in body fluids that have not been previously considered and the second is the notion that neurodegenerative disorders may not require either a single biomarker or two biomarkers considered in isolation, but rather a combination of parameters. Our approach also aims to achieve a robust method that can be used by artificial intelligence in an unbiased manner. Artificial intelligence requires algorithms capable of analysing large amounts of patient data from various sources, but also the ability to make predictions based on the integration of data sets. Any advances in identifying integration patterns and correlating patient data sets will be crucial for personalized diagnosis, prognosis, and treatment selection.

The analysis was performed using metabolomics data from the AH of healthy controls, open-angle glaucoma patients, and type 2 diabetes patients. We implemented an intelligent-based classifier using linear discriminant analysis (LDA) as the core technique as a proof of concept to implement more advanced methods to increase the discrimination power, if necessary, in forthcoming studies. Using the leave-one-out approach, we evaluated the overall accuracy, sensitivity, specificity, positive predictive value (PPV), and negative predictive value (NPV) of this novel approach that provides a tool to move towards truly personalized medicine.

## 2. Materials and Methods

### 2.1. Subjects

A total of 46 samples of AH were collected; 31 samples from healthy controls (15 men and 16 women, mean age 56, range 24–76), 8 samples from open-angle glaucoma patients (4 men and 4 women, mean age 67.5, range 51–81), and 7 samples from type 2 diabetes patients (4 men and 3 women, mean age 72, range 65–76). None of the individuals had previously undergone eye surgery. By the characteristics of people undergoing refractive surgery, from which AH can be obtained, controls are, in proportion, higher in number than patients with open-angle glaucoma or type 2 diabetes; they are also younger.

We compared groups in terms of age. The F index value is 6.708 and the *p* value is 0.002913, so the result is significant at *p* < 0.05, which means that there are age differences between the groups. For this reason, we repeated the analysis excluding younger controls, with similar results as detailed in Section 3. Regarding other characteristics, none of the individuals reported any kidney disease, and creatinine and urea plasma levels were within reference values. None of the diabetic patients showed any sign of diabetic retinopathy.

The study was evaluated by the “Comitè d’Ètica de la Investigació de les Illes Balears (CEI-IB)” and deemed not to require ethics approval. Samples are considered waste and no data on patient identification (neither name, address nor ID numbers) are available to experimenters.

### 2.2. Metabolomics

The AbsoluteIDQ™ p180 Kit (Biocrates Life Sciences, Innsbruck, Austria), which can determine 188 metabolites, from biogenic amines, amino acids, hexoses, phospho- and sphingolipids to acyl-carnitines was used. Individual metabolites may be found in www.biocrates.com/products/research-products/absoluteidq-p180-kit (Accessed on 1 July 2022). Up to 30 μL of AH were plated in each well. The initial sample processing was as indicated by the manufacturer. Afterwards, derivatized samples were analysed in the AB Sciex 6500 QTRAP MS/MS mass spectrometer (AB Sciex LLC, Framingham, MA, USA) coupled to an Agilent 1290 Infinity UHPLC system (Agilent, Santa Clara, CA, USA). Data analysis was performed using Analyst (v. 1.7.3) and the MetIDQ™ (v. 5.5.4-DB100 Boron-2623) software.

### 2.3. Data Compilation and Computational Analyses

Data (concentration of every metabolite in each sample) used in the present study are available in previously published papers [5,14]. We used R 4.3.0 software to conduct linear discriminant analysis (LDA) with leave-one-out cross-validation to control for overfitting. We used the CAR and MASS libraries as R scripts programmed by the authors, details can be provided by request to the first author (DBC). In addition, further details of the implementation and specific analyses are provided in the Section 3.

## 3. Results

The main objective of this study was to test whether the combination of metabolomics and advanced computational tools comprising statistical methods, machine learning, and/or artificial intelligence approaches can lead to the development of novel diagnostic tools for various diseases. The standard pipeline of an intelligent-based classifier consists of three well-differentiated layers: the input layer, the hidden layer, and the output layer. The input layer depends on the data and consists of 188 metabolites. The hidden layer may include one or several methods ranging from simple regression models to more complex deep learning methods. To this end, we designed a novel pipeline using LDA as a core technique in the hidden layer, a layer that, in turn, consisted of two learning steps. The output layer corresponds to the different classes, in our case, the three groups: glaucoma, (type 2) diabetes, and control. Figure 1 illustrates a generic diagram and the scheme implemented in this article.

### 3.1. Linear Discrimination Analysis (LDA) Considering All Metabolites

The first objective was to describe the results obtained by implementing the hidden layer with LDA as a core platform, a layer consisting of two steps (Figure 1). Subsequently, the descriptive statistics and correlations needed to validate the approach are described and evaluated.

To begin with, we ran an LDA for every metabolite and computed the overall accuracy, sensitivity, specificity, PPV, and NPV considering all subjects using a leave-one-out cross-validation strategy to control for overfitting. Phosphatidylcholine diacyl C42:6 (PC aa C42:6) was the metabolite with the best discriminatory power (Figure 2 and Figure 3). Then, we ran LDAs with a combination of two, three, four, five, six, seven, and eight metabolites, which were randomly selected from the original pool of 188 metabolites. Looking for the maximum leave-one-out accuracy we found that the pair of metabolites with the best performance were acetyl-ornithine (Ac-Orn) and PC aa C42:6. In fact, the maximum discrimination accuracy was obtained with four metabolites, but addition of a fifth or a sixth, etc., did not lead to better results. In other words, adding more compounds would not provide superior results. The four metabolites were C3 acyl-carnitine (C3), C3-DC (C4-OH) acyl-carnitine (C3-DC (C4-OH)), Ac-Orn, and PC aa C42:6, for which the peak performance is around 95% (Figure 2). In subsequent computations, we only considered these four metabolites.

#### 3.1.1. Descriptive Statistics: Boxplots

We then evaluated descriptive statistics for each of the selected metabolites. Boxplots showing the concentration of the four selected metabolites (C3, C3-DC (C4-OH), Ac-Orn, and PC aa C42:6) provide information as to why these metabolites are the most discriminative. The four boxplots for each of the selected metabolites showing the differences among groups are depicted in Figure 3. While the metabolite PC aa C42:6 allows discrimination for glaucoma, Ac-Orn superbly discriminates diabetes from controls. These two patterns and the discriminative power of C3 and C3-DC (C4-OH), allow for a 95% discrimination accuracy among groups.

#### 3.1.2. Correlation Analysis

In addition, relevant information can be obtained by evaluating whether the four metabolites: C3, C3-DC (C4-OH), Ac-Orn, and PC aa C42:6 are correlated (Table 1). The 16 scatterplots showing the relationship between each pair of metabolites is depicted in Appendix A, which shows the values corresponding to one metabolite of a given pair of metabolites on the horizontal axis, and the values corresponding to the second metabolite on the vertical axis. In each graph every point corresponds to data from a sample (control, glaucoma, or diabetes).

One important component of this type of scatterplots in Appendix A is the direction of the relationship between the two variables. We observe a positive association between each pair of metabolites with no exception. In other words, above-average values of one metabolite tend to accompany above-average values of the other metabolite, and below-average values also tend to occur together. We conclude that this general trend constitutes a pattern that allows an adequate classification of everyone in their respective group. Another important component is the geometric form of the relationship between two metabolites. In general, the points on the scatterplots can fit on a straight line with a few exceptions. For example, the relation between C3 and C3-DC (C4-OH) seems to be quadratic. Finally, another critical component is the strength of the relationship between two variables. The slope provides information on the strength in such a way that measures the degree of variation of one variable when the second increases. Thus, we observe a strong linear relationship between metabolites C3 and PC aa C42:6, which is very relevant as the product of the two metabolites will be selected in the second step to achieve 100% accuracy.

We observed a high degree of correlation among the metabolites with all pair-wise correlations being significant. The statistical significance and high degree of correlation within the four selected metabolites suggests a metabolic signature (Figure 3 and Table 1).

### 3.2. Linear Discrimination Analysis (LDA) Considering the Most Discriminative Metabolites and Selecting an Optimal Non-Linear Combination

In a second step, high-order terms were searched using the concentration of the previously selected four metabolites: C3, C3-DC (C4-OH), Ac-Orn, and PC aa C42:6. We went up to third-order terms. In other words, we considered linear, quadratic, and cubic terms plus double and triple interactions. We proceed in the same way as in the previous LDA step, i.e., the number of variables was increased until achieving the maximum leave-one-out accuracy. The maximum leave-one-out accuracy was reached with six variables and including more variables did not change the accuracy. We found significantly relevant the level of Ac-Orn and the squared concentration of acyl ornithine: Ac-Orn^2^ (Figure 4). The other two relevant parameters were the levels of C3 and PC aa C42:6. More specifically, we found that C3*Ac-Orn, C3*PC aa C42:6, C3^2^*Ac-Orn, and C3*Ac-Orn^2^ parameters are relevant (Figure 3) and that the level of C3-DC (C4-OH) can be omitted. With these six variables resulting from knowing the levels of three metabolites, we achieved a 100% leave-one-out accuracy. Including more variables could not increase the accuracy, which was already 100% with the ones selected.

#### 3.2.1. Descriptive Statistics: Boxplots

We can use boxplots as before to illustrate the discriminative power of each of the selected variables. Indeed, we can reasonably interpret the behaviour of all the six variables. They offer different patterns of discrimination that, when combined, allow for 100% classification accuracy (Figure 5).

Thus, the variable C3*PC aa C42:6 allows for discriminating glaucoma from the rest. The variable Ac-Orn and the variable Ac-Orn^2^ allow for differentiating diabetes from controls. The variable C3*Ac-Orn and the variable C3*Ac-Orn^2^ seem also to contribute to this classification but also helps to distinguish controls from the rest. Finally, the variable C3^2^*Ac-Orn aims substantially to discriminate controls from the rest.

Importantly, by using all variables together and not just one, we can classify all subjects into their groups, motivating the use of multivariate techniques in metabolomics.

#### 3.2.2. Correlation Analysis

As in the data earlier presented, a positive association between each pair of variables was noted. In general, the points on the scatterplots can fit on a straight line with a few exceptions (Appendix A). For example, a curvilinear relationship is observed between the variable C3*PC aa C42:6 and the variable C3*Ac-Orn. Importantly, regarding the strength of the relationship between variables, we observe strong linear relationships between variables that increase the classification performance. Thus, we observe the strongest linear relationship between the variable C3∗Ac-Orn and the variable C3^2^*Ac-Orn (Pearson coefficient = 0.8972, *p*-value < 0.00001). It is worthwhile to stress that C3*Ac-Orn is a selected variable of the model, and C3^2^*Ac-Orn is the same variable multiplied by C3; if C3 was a constant, the correlation would be exactly 1.

Relevant information can be obtained by evaluating whether the six selected variables, which allowed a 100% discrimination power, were correlated (Table 2). We observe a high degree of correlation among variables, with almost all pair-wise correlations being significant. The statistical significance and high degree of correlation within the six selected variables suggests a more accurate (non-linear) metabolic signature (Figure 5 and Table 2).

The 36 scatterplots showing the relationship between each pair of variables is depicted in Appendix A. The six variables are: Ac-Orn, Ac-Orn^2^, C3*Ac-Orn, C3*PC aa C42:6, C3^2^*Ac-Orn, and C3*Ac-Orn^2^. In each scatterplot in Appendix A, the values corresponding to one variable of a given pair of variables appear on the horizontal axis, and the values corresponding to the second appear on the vertical axis. In each graph every point corresponds to data from a sample (control, glaucoma, or diabetes).

Taken together, it appears that the non-linear pooling of the concentration of these metabolites allows all the individuals to be correctly classified. In future studies, we anticipate that the non-linear pooling feature will be of extreme importance for optimal performance of AI-based diagnostic tools.

#### 3.2.3. Non-Linear Method to Achieve 100% Accuracy

Intelligent techniques hold promise for disease diagnosis, but the underlying substrate is currently missing [15]. Intelligent methods use learning algorithms ranging from very simple linear regression to LDA approaches and complex deep learning tools. They work on training and testing data sets so that the system can “detect” a disease to make an early diagnosis. Our intelligent-based method uses LDA as a core algorithm and consists of considering the metabolites useful for discrimination among the two diseases and healthy controls and providing a diagnosis. As expected, the method applied to all samples, cases, and controls, led to the confusion matrix in Table 3, which shows full success in stratification. In fact, the overall accuracy, sensitivity, specificity, PPV, and NPV are 100%.

As reported in the Section 2, there was a statistical difference between the groups in terms of age, so we reran the analyses excluding the younger controls. The reduced data set consisted of 20 controls, 8 glaucoma and 7 diabetes. Importantly, the analysis, after excluding the younger controls, yielded similar results (Table 4). Essentially, after excluding the young control cases, what happens is that one control is misplaced as glaucoma; none of the cases, neither glaucoma nor diabetes, appear as controls using the new equations obtained.

We also computed the a posteriori probability for each subject. A posteriori probability is a probability obtained from Bayesian reasoning. We can initially assign a probability that an individual belongs to a group based on vague initial information. For example, if there are three possibilities, we can consider the probability of a subject belonging to a group. This allows us to ‘assume’ or ‘suppose’ a prior probability and then obtain a posteriori probability from handling the data using the developed algorithm. Such “a posteriori” probability, if the algorithm works, should be close to one. Figure 6 illustrates the a posteriori probability calculated for each subject.

#### 3.2.4. 2D Map Space Reflecting Position of Controls, Glaucoma Patients, and Type 2 Diabetes Patients

To allow a graphical depiction of the regions occupied by the three groups: control, glaucoma, and diabetes, two discriminant functions are required, i.e., total number of groups minus 1. For the underlying geometry the Mahalanobis distance was considered.

The two discriminant functions LD1 and LD2 are:LD1 = 10.221 × **Ac-Orn** − 6.133 × **Ac-Orn^2^** − 63.218 × **C3*****Ac-Orn** + 345.456 ×**C3*****PC aa C42:6** + 30.440 × **C3^2^*****Ac-Orn** + 27.188 × **C3*****Ac-Orn^2^**.(1)
LD2 = 4.512 × **Ac-Orn** − 3.491 × **Ac-Orn^2^** − 47.064 × **C3*****Ac-Orn** − 28.857 × **C3*****PC aa C42:6** + 45.883054 × **C3^2^*****Ac-Orn** + 16.104838 × **C3*****Ac-Orn^2^**.(2)

These equations come from applying the LDA in the space generated by the six variables considered.

The discriminant regions constrained by the LDA are illustrated in Figure 7; essentially, they are the cake-like pieces that delimit each group. Each region is determined by the closest points to the centre of mass of each group. The criterion is purely geometric, coincident with the maximum likelihood, under the hypothesis of multivariate normality. The three boundaries among the three groups are optimal and guarantee 100% success in diagnosis using the above-described protocol. Each dot in Figure 7 corresponds to the data, LD1 and LD2, from the sample of an individual.

## 4. Discussion

Historically, biomarkers still in use today were developed using clinical chemistry approaches and determining the concentration of small compounds in urine and serum/plasma. The search for valuable biomarkers to detect disease and assess therapy or disease progression has recently relied more on genomic and proteomic techniques than metabolomic approaches. However, this trend may soon change in favour of metabolomics due to the recent possibility of determining hundreds of compounds with high precision using a few microliters of sample. This opens a new window of opportunity to identify small-molecule biomarkers for virtually any disease.

Almost any critical metabolite that is overrepresented or underrepresented in disease conditions may be among those 200–500 small molecules whose determination in body fluids is now possible via high-throughput metabolomics. Due to the potency of newly developed mass spectrometry-based methodology, it is possible to simultaneously measure, in microliters of samples placed in 96-well plates, a variety of lipophilic and hydrophilic compounds, including amino acids, biogenic amines, glycerophospholipids, sphingomyelins, and acyl-carnitines. Indeed, with the advent of metabolomics, the literature concerning studies carried out using samples of patients with different diseases ranging from cardiovascular to neurological disorders is growing exponentially [16,17,18,19].

The altered concentration of some metabolites found in the AH of patients in comparison to healthy individuals is providing valuable insights into the pathophysiological mechanisms of multiple diseases, including, but not limited to, open-angle glaucoma and type 2 diabetes [5,14], but a diagnostic tool is missing. Our laboratory has recently provided metabolomics-based evidence of altered arginine metabolism as a principal factor in Parkinson’s disease [20]. Usually, the studies give insight into the pathophysiological mechanisms while being unable to propose useful/promising biomarkers. Diagnostic tools are lacking because a significant drawback is the procedure for selecting disease-specific biomarkers capable of addressing disease progression, which is specifically challenging in chronic disease.

The present study moves towards combining metabolomics data with intelligent algorithms to improve the discrimination power of modern, i.e., metabolomics-based techniques in clinical chemistry research. The paradigm supported here demonstrates the utility of combining metabolomics with advanced computational approaches to distinguish between patients with open-angle glaucoma, type 2 diabetes, and healthy controls. The novel method uses LDA as the learning algorithm. The scheme is based on the standard intelligent-based pipeline, and the hidden layer consists of two learning steps. First, we select the most discriminative metabolites among them. Then, we combine the significant ones to mathematically generate higher-order terms to classify people into glaucoma, diabetes, and control with 100% accuracy.

After executing the full protocol, the final three selected metabolites are Ac-Orn amino acid, C3 acyl-carnitine, and PC aa C42:6 phosphatidylcholine. The resulting six classification variables are Ac-Orn, Ac-Orn^2^, C3*Ac-Orn, C3*PC aa C42:6, C3^2^*Ac-Orn, and C3*Ac-Orn^2^. Interestingly, the “intelligent” algorithm selects metabolites from different biochemical families resulting in a method that fits all subjects regardless of their pathological condition. Very relevant is that 100% success is achieved with only three metabolite concentrations resulting into 6 variables consisting of non-linear combinations.

In our previous publications concerning the composition of AH, we were interested in looking for differentially concentrated metabolites and why they could give insight into disease mechanism. We discovered that glutamine, kynurenine, acyl-carnitine and lysophosphatidylcholine levels are altered in the AH of glaucoma patients [14], that biogenic amines are differentially concentrated in the AH of type 2 diabetes patients [5] and that the metabolism of arginine is altered as deduced from the composition of the AH of Parkinson’s disease patients [20]. In the present study, we use the data from previous studies to develop a novel methodology for disease diagnosis and management considering the perspective of personalized medicine. Accordingly, the selected metabolites may not have any relevant function at the level of either pathophysiology or recovery from disease. This lack of concern for the causes of altered metabolite levels may seem limiting, but it leads to unbiased analysis. The enormous task of considering all possibilities is necessary to select the simplest but most successful (linear or non-linear) combination of metabolites, but once the final algorithm is obtained, the diagnostic/prognostic tools are ready to use. Furthermore, since only 3–5 metabolites are required, the robustness of the method is not linked to enormous costs when translated to the clinical practice, but to the determination of the level of those 3–5 molecules in the ad hoc body fluid.

The main limitation of the study is the impossibility of external validation of the approach. This is due to the uniqueness of the samples, constituted by AH, which in fact cannot be used for routine diagnosis/prognosis. The method consists of a proof-of-concept on whether the linear and/or non-linear combination of the level of a few metabolites can serve for diagnosis/prognosis using body fluids such as plasma, urine, sweat, or cerebrospinal fluid. Another limitation is the reduced sample size and the unequal distribution of ages between the control group and the two patient groups. However, by implementing cross-validation with a leave-one-out strategy, we demonstrate that the method is robust, that is, that, despite all limitations, it provides excellent discriminative power. Reducing the control group to cancel out differences between groups in terms of age leads to “almost” the same classification accuracy, except for one control. Following the use of, for example, serum and external validation using enough patients and age-matched controls, our method will hopefully be translatable to clinical practice in a precision medicine context.

In line with this study, a few papers have recently been divulged on implementing tools for diagnosing glaucoma [21,22] and type 2 diabetes [23,24]. In addition, two patent applications have recently been filed by colleagues from France [25] and Spain [26] to reveal metabolomic signatures for glaucoma; the latter application seems ready for commercialization. However, to date, we have not found any report of classifying diseases together or in conjunction with intelligent-based methods.

It is imperative to implement clinical tools that help doctors in their practice with a special focus on precision diagnosis and personalized medicine. Our results are auspicious, and the procedure we present may pave the way for building an intelligent-based diagnostic and assessment tool with algorithms that can be further used in clinical research. The possibility of using the tool to evaluate the progression of the disease and/or evaluate the efficacy of new therapeutic routes is also relevant. The success of this project will foster the integration of two emerging technologies, metabolomics, and advanced computational methods, which have yet to be fully harnessed together.

## 5. Conclusions

Our results open the possibility to devise a method to identify a number of clinical chemistry parameters in human body fluid, that in combination could lead to the ability to differentiate between diseases. Equations such as those here presented to differentiate between open-angle glaucoma and type 2 diabetes using samples from AH of the eye, can be implemented using any fluid for which reliable levels of metabolites can be obtained. By analysing such levels in samples from individuals with different diseases, it would be possible to select a few of them and devise an equation that would lead to unbiased and reliable diagnosis. Once the 3–5 parameters are selected, the method would be cost-effective because it would consist of the determination the level of those 3–5 molecules in the ad hoc body fluid. Hence, the method here proposed can be scaled up using data from several diseases with the goal of building a “digital assistant” capable of diagnosing, from serum, urine, sweat, and/or cerebrospinal fluid, any disease, with high precision and allowing patient stratification, and assessment of therapy efficacy and disease progression. The strategy here put forward, in our opinion, would be valuable in a personalized/precision medicine context.

## Figures and Tables

**Figure 1 metabolites-14-00149-f001:**
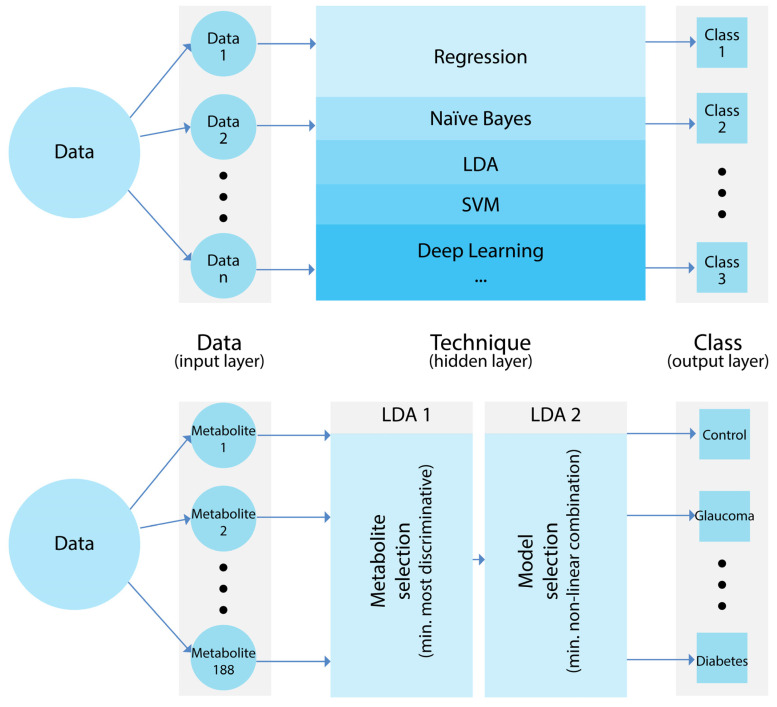
Standard pipeline of an intelligent-based classifier (**top**) and our intelligent-based classifier using LDA as a core platform (**bottom**).

**Figure 2 metabolites-14-00149-f002:**
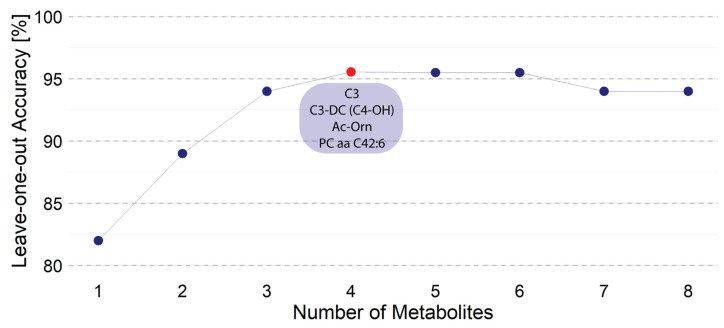
Accuracy through leave-one-out cross-validation. All possible combinations were tested selecting one metabolite and adding a second, third …; i.e., adding, one by one, metabolites to the classifier. The image shows the best result obtained from all possible combinations. With four molecules (red dot) the peak accuracy was circa 95%; the rest of combinations are represented with blue dots. Adding more metabolites to the four indicated metabolites did not improve precision.

**Figure 3 metabolites-14-00149-f003:**
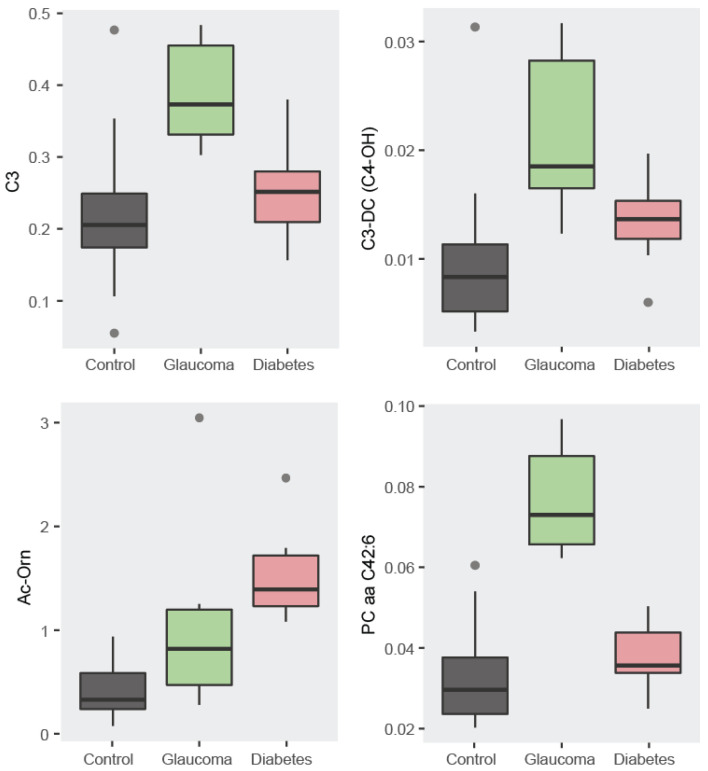
Boxplots for the four selected metabolites. For each of the four metabolites there are concentration differences between the groups that improve classification accuracy by taking them all together instead of using a single metabolite.

**Figure 4 metabolites-14-00149-f004:**
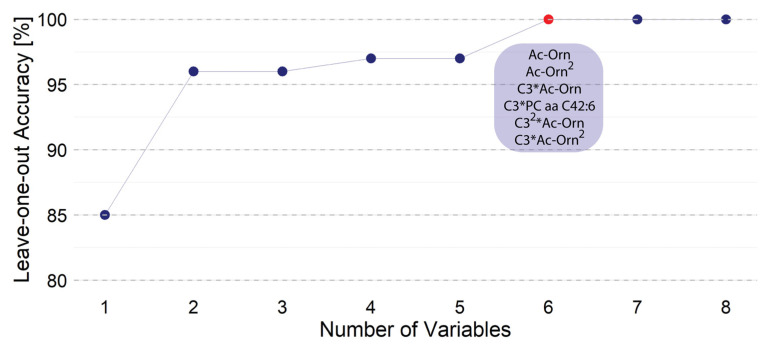
Accuracy through leave-one-out cross-validation considering the most discriminative metabolites and selecting the minimum number for non-linear relationship. With six terms (red dot) in the non-linear equation the peak accuracy was 100%. Upon considering all possible combinations (blue dots), six was the minimum number of variables providing full (100%) accuracy.

**Figure 5 metabolites-14-00149-f005:**
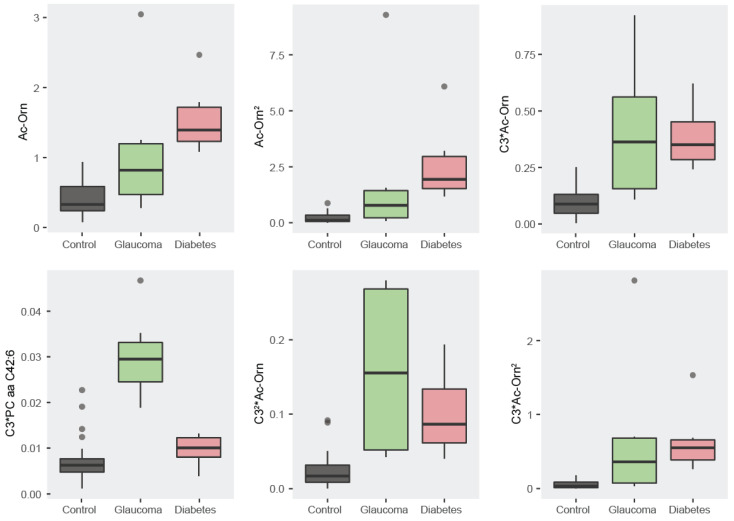
Boxplots for the six selected variables. On each of the six variables, there are substantial differences between the groups that collectively contribute to discrimination.

**Figure 6 metabolites-14-00149-f006:**
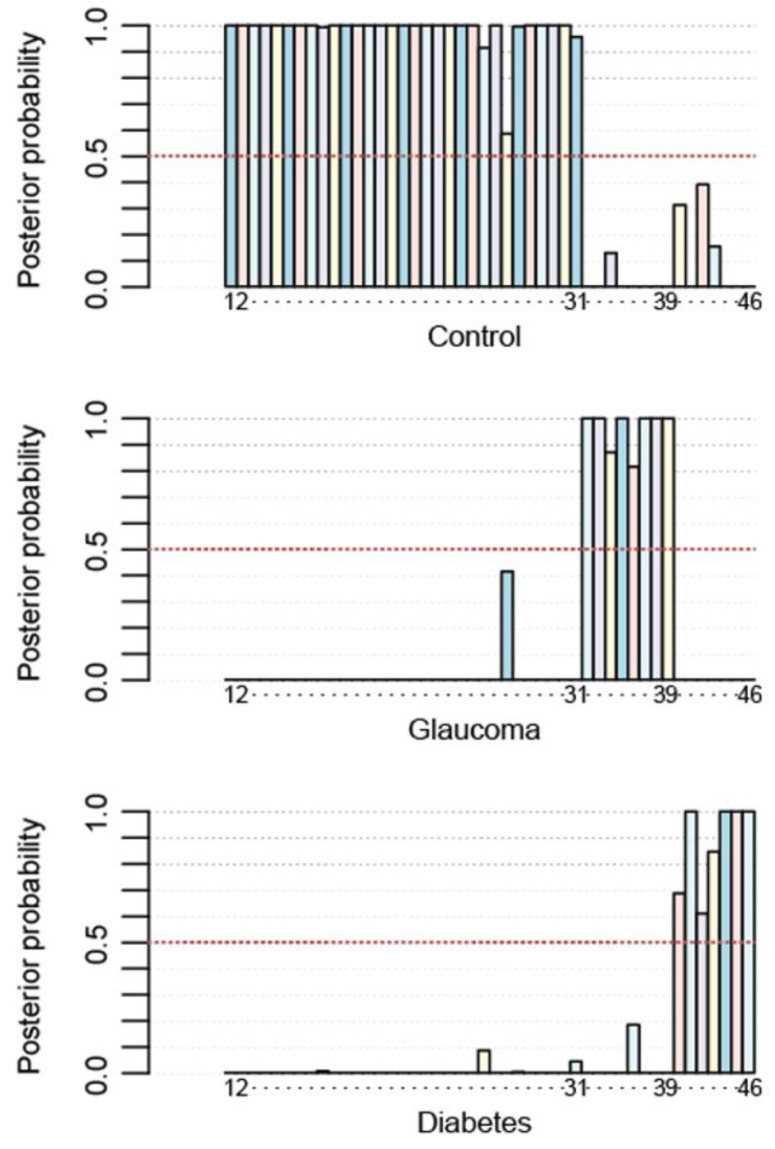
A posteriori probability for each subject. Posterior probabilities of individuals being classified into the control, glaucoma, and diabetes groups, respectively, using the LDA learned with leave-one-out cross-validation. Each bar represents a subject, depicted with a different colour. The overall performance of the LDA is 100% (data of members in a given group have a greater posterior probability of 0.5 for the graph in the corresponding group).

**Figure 7 metabolites-14-00149-f007:**
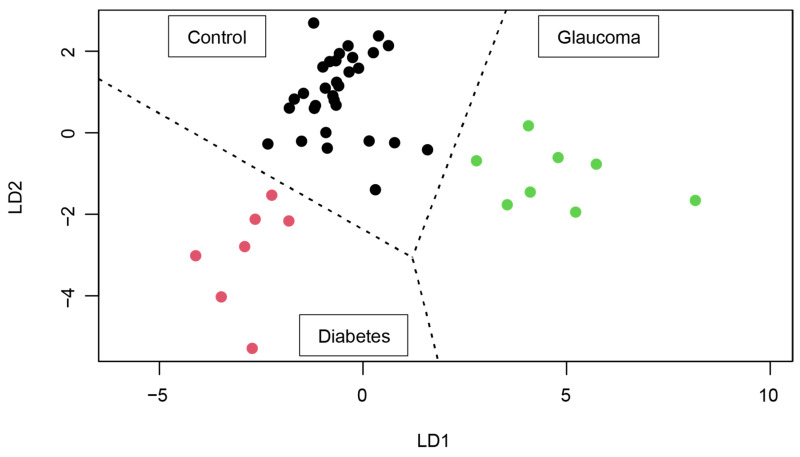
Discriminant regions. Note that the LD1 values and the LD2 values are non-linear combinations of the variables (see Equations (1) and (2)), therefore they can be calculated for any individual. Each subject in the data appears as a point, whose group depends on a colour: control (black), glaucoma (green), and diabetes (magenta).

**Table 1 metabolites-14-00149-t001:** **Correlation between the four selected metabolites**. The Pearson coefficient is shown for pair-wise correlations. A total correlation would be defined by coefficient = 1. The asterisk denotes *p*-value < 0.05.

C3	0.4921 *	0.3685 *	0.5109 *
	C3-DC (C4-OH)	0.3018 *	0.4539 *
		Ac-Orn	0.3703 *
			PC aa C42:6

**Table 2 metabolites-14-00149-t002:** **Correlation between the six selected variables**. The Pearson coefficient is shown for pair-wise correlations. A total correlation would be defined by coefficient = 1. The asterisk denotes uncorrected *p*-value < 0.05.

Ac-Orn	0.8575 *	0.3760 *	0.2715	0.1843	0.1986
	Ac-Orn^2^	0.3719 *	0.1813	0.1710	0.3177 *
		C3*Ac-Orn	0.7258 *	0.8972 *	0.7186 *
			C3*PC aa C42:6	0.7808 *	0.4784 *
				C3^2^*Ac-Orn	0.7293 *
					C3*Ac-Orn^2^

**Table 3 metabolites-14-00149-t003:** **Confusion matrix.** Real means the diagnostic received by the clinician and test means the result of the stratification made upon applying the equations here devised.

	Real	Control	Glaucoma	Diabetes
Test				
Control		31	0	0
Glaucoma		0	8	0
Diabetes		0	0	7

**Table 4 metabolites-14-00149-t004:** **Confusion matrix for the analysis resulting from excluding the youngest within the control group.** Real means the diagnostic received by the clinician and test means the result of the stratification made upon applying the equations here devised.

	Real	Control	Glaucoma	Diabetes
Test				
Control		19	0	1
Glaucoma		0	8	0
Diabetes		0	0	7

## Data Availability

Data and resources may be provided by the corresponding (RF) or the first (DBC) authors upon reasonable request. All raw data used in this study are available as Appendix A of previously published papers [5,14,20].

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
