# Peer review of "Advancing Personalized Medicine by Analytical Means: Selection of Three Metabolites That Allows Discrimination between Glaucoma, Diabetes, and Controls"

_metabolites, 2024, doi:10.3390/metabo14030149_

Round 1
Reviewer 1 Report
Comments and Suggestions for Authors
The article titled "Advancing personalized medicine by analytical means: selection of three metabolites that allows discrimination between glaucoma, diabetes, and controls" is interesting and presents an interesting approach to the diagnosis of selected diseases. However, I ask that the authors address the following questions or suggestions.
1. The number of patients varies strongly between groups (31 vs 8 vs 7). Does the age of the patients appear to be significantly different between the study groups? What is the exact p-value of such a comparison?
2. The control group is very interesting. For what reason was an ophthalmic operation performed among these patients, during which the eye's aqueous fluid was collected?
3. The entire inference in the paper is based on studying metabolite levels in the ocular aqueous fluid. Do the authors think that the methods presented here can be applied to other materials, e.g. blood serum, tears? Moreover, I believe that the authors should indicate in the abstract and conclusions that the test material was the aqueous fluid of the eye.
4. The authors do not state what are the limitations of the study that was conducted.
5. The authors do not provide information on what role artificial intelligence played in the preparation of the manuscript.
6. The authors do not state what packages they used in R.
Comments on the Quality of English LanguageThere are spelling errors in some words.
Author Response
The article titled "Advancing personalized medicine by analytical means: selection of three metabolites that allows discrimination between glaucoma, diabetes, and controls" is interesting and presents an interesting approach to the diagnosis of selected diseases. However, I ask that the authors address the following questions or suggestions.
- The number of patients varies strongly between groups (31 vs 8 vs 7). Does the age of the patients appear to be significantly different between the study groups? What is the exact p-value of such a comparison?
Answer. Thanks for the comments. By the characteristics of people undergoing surgery, refractive correction. from which aqueous humour can be obtained, controls are, in proportion, higher in number than patients of type 2 diabetes or open-angle glaucoma. Also, the number of controls is much higher than the number of patients. We tested for the statistical significance among groups in terms of age. The F-ratio value is 6.708, and the p-value is 0.0029, so that the result is significant (p < 0.05). However, we have redone the analysis excluding younger controls with similar results. The reduced dataset consisted of 20 controls, 8 glaucoma, and 7 diabetes. The results are included in the revised version of the paper (Section 3.2.3. Non-linear method to achieve 100% accuracy of the updated manuscript). Essentially what happens is that only one control is misplaced as glaucoma; none of the cases, neither glaucoma nor diabetes, appear as controls using the new equations obtained after elimination of young control cases.
- The control group is very interesting. For what reason was an ophthalmic operation performed among these patients, during which the eye's aqueous fluid was collected?
Answer. Indeed, this was one of the important issues that prompted us to start with this type of samples. It is refractive correction, i.e. no ocular-related disease that could impact on the composition of the HA. This information is now included in the revised version of the paper.
- The entire inference in the paper is based on studying metabolite levels in the ocular aqueous fluid. Do the authors think that the methods presented here can be applied to other materials, e.g. blood serum, tears? Moreover, I believe that the authors should indicate in the abstract and conclusions that the test material was the aqueous fluid of the eye.
Answer. Thanks for the comment. Yes, this is the whole idea to implement a procedure that could be used as diagnostic/prognostic tool using blood, etc. Aqueous humour cannot be used for diagnosis/prognosis due to ethical issues; we started to use this fluid to try to find metabolic alterations (ideally in neurological diseases) that were not observed using another type of sample. In summary, yes we want to go, next, to more readily available body fluids.
- The authors do not state what are the limitations of the study that was conducted.
Answer. Limitations have been included in the revised version of the paper. Apart from the sample size and the younger control group versus the glaucoma and diabetes groups, limitations include the unavailability of checking the models on an external validation group. However, we coped with this limitation by implementing a cross-validation with a leave-one-out strategy. Also, reducing the control group to achieve no differences among groups in terms of age and having “almost” the same classification accuracy except for one control confirms that our models are robust and suitable for clinical practice.
- The authors do not provide information on what role artificial intelligence played in the preparation of the manuscript.
Answer. Thanks for raising the issue. Our idea is that AI could use our procedure to make non-biased diagnosis using data from blood, etc. tests. We have rewritten some sentences to make this clearer.
- The authors do not state what packages they used in R.
Answer. The packages used have been detailed in the revised version of the manuscript. We have used the CAR and MASS libraries as R scripts programmed by the authors, who are happy to share by request.
Modifications are in red lettering. Minor changes are not highlighted.
Reviewer 2 Report
Comments and Suggestions for Authors
This paper took the concentration of 188 compounds measured in the aqueous humor of patients and controls, and linear (LDA) and non-linear discriminant analysis was used to identify the right set of compounds to select four compounds that allow distinguishing between open-angle glaucoma patients, type 2 diabetes patients, and healthy controls through an intelligence-based method. Results showed that LDA selected acetyl-ornithine (Ac-Orn), C3 acyl-carnitine (C3), diacyl C42:6 phosphatidylcholine (PC aa C42:6), and C3-DC (C4-OH) acyl-car-nitine (C3-DC (C4-OH)) with 95% discriminative success and 100% success was obtained with a nonlinear combination of the concentration of the four compounds. Conclusions indicates that the proposed approach makes possible an intelligent device capable of diagnosing diseases with high precision also allowing patient stratification and disease progression assessment.
The paper organized well and the proposed method sounds feasible. The paper is ready to publication except the font size in the figures which is a little bit lager than the text.
Author Response
This paper took the concentration of 188 compounds measured in the aqueous humor of patients and controls, and linear (LDA) and non-linear discriminant analysis was used to identify the right set of compounds to select four compounds that allow distinguishing between open-angle glaucoma patients, type 2 diabetes patients, and healthy controls through an intelligence-based method. Results showed that LDA selected acetyl-ornithine (Ac-Orn), C3 acyl-carnitine (C3), diacyl C42:6 phosphatidylcholine (PC aa C42:6), and C3-DC (C4-OH) acyl-car-nitine (C3-DC (C4-OH)) with 95% discriminative success and 100% success was obtained with a nonlinear combination of the concentration of the four compounds. Conclusions indicates that the proposed approach makes possible an intelligent device capable of diagnosing diseases with high precision also allowing patient stratification and disease progression assessment.
The paper organized well and the proposed method sounds feasible. The paper is ready to publication except the font size in the figures which is a little bit lager than the text.
Answer. Thanks for the positive comments. Font size in figure legends has been made smaller than in the text.
Modifications are in red lettering. Minor changes are not highlighted.
Reviewer 3 Report
Comments and Suggestions for Authors
Bernal-Casas et al.'s manuscript described an intelligence-based method for selecting metabolites that can distinguish between glaucoma patients, diabetes patients, and healthy controls. The study demonstrates that by using LDA and six variables, which are non-linear combinations of three metabolites (acetyl-ornithine, acyl-carnitine C3, and phosphatidylcholine diacyl C42:6), it is possible to achieve 100% accuracy in correctly classifying individuals into glaucoma, diabetes, and control groups. The study concludes that this approach can lead to a novel diagnostic tool that requires a minimum number of significant metabolites while maintaining high predictive values. The significance of this research lies in its potential to advance personalized medicine by using a combination of metabolites rather than relying on single biomarkers.
Comments:
1. Line 93-96: Please indicate what tissue type was collected for metabolomics.
2. Provide more details about the six variables used in the classifier. A boxplot displaying the concentration of four metabolites (C3, C3-DC (C4-OH), Ac-Orn, and PC aa C42:6) can provide insights into why these metabolites are the most discriminative.
3. The sample size in this study was too small. Consider conducting further validation studies using larger and more diverse datasets. This would help assess the robustness and reliability of the proposed method across different populations and settings. While the study focuses on glaucoma and diabetes, it would be interesting to explore if the same approach can be applied to other diseases with similar accuracy.
4. Discuss the potential limitations and challenges of implementing the proposed method in a clinical setting. Consider factors such as cost, accessibility of metabolite measurements, and integration with existing diagnostic tools.
Author Response
Bernal-Casas et al.'s manuscript described an intelligence-based method for selecting metabolites that can distinguish between glaucoma patients, diabetes patients, and healthy controls. The study demonstrates that by using LDA and six variables, which are non-linear combinations of three metabolites (acetyl-ornithine, acyl-carnitine C3, and phosphatidylcholine diacyl C42:6), it is possible to achieve 100% accuracy in correctly classifying individuals into glaucoma, diabetes, and control groups. The study concludes that this approach can lead to a novel diagnostic tool that requires a minimum number of significant metabolites while maintaining high predictive values. The significance of this research lies in its potential to advance personalized medicine by using a combination of metabolites rather than relying on single biomarkers.
Comments:
- Line 93-96: Please indicate what tissue type was collected for metabolomics.
Answer. Thanks for noticing. The type of sample is specified in this part of the manuscript.
- Provide more details about the six variables used in the classifier. A boxplot displaying the concentration of four metabolites (C3, C3-DC (C4-OH), Ac-Orn, and PC aa C42:6) can provide insights into why these metabolites are the most discriminative.
Answer. Thanks for the suggestion.
We have included boxplots displaying the concentration of the four selected metabolites in the first step of the algorithm (Figure 3 of the revised manuscript). In addition, we have also attached boxplots displaying the concentration of the six selected variables in the second step of the algorithm (Figure 5 of the revised document). With these plots and the ad hoc explanations, the manuscript has substantially improved in quality, as we can visually explain why it is needed the discrimination power of a “combination of variables”, a single variable would not be sufficient. We have included both descriptions in the updated version. In addition, we have also included a correlation analysis of the selected metabolites.
- The sample size in this study was too small. Consider conducting further validation studies using larger and more diverse datasets. This would help assess the robustness and reliability of the proposed method across different populations and settings. While the study focuses on glaucoma and diabetes, it would be interesting to explore if the same approach can be applied to other diseases with similar accuracy.
Answer. Unfortunately increasing the sample is not possible. What we believe is one of the strengths of our approach is that with these number of samples (small due to the limitations of having this type of samples) the results are robust. This topic has been addressed in response to the point 4 and of another reviewer that suggested to specify the limitations of our study. Despite limitations the method provided appears to be quite robust.
- Discuss the potential limitations and challenges of implementing the proposed method in a clinical setting. Consider factors such as cost, accessibility of metabolite measurements, and integration with existing diagnostic tools.
Answer. Thanks for the suggestion. Limitations have been added to in the revised version. The whole idea to implement a procedure that could be used as diagnostic/prognostic tool using blood, etc. Aqueous humour cannot be used for diagnosis/prognosis due to ethical issues; we started to use this fluid to try to find metabolic alterations (ideally in neurological diseases) that were not observed using another type of sample. In summary, we developed a method using the data we had from the sample we had but we are pretty confident that it may be used in clinical settings and we are working on this via novel studies and via collaborations with clinicians (everything is quite slow due to ethical issues and the need to protocol approvals)
Modifications are in red lettering. Minor changes are not highlighted.
Reviewer 4 Report
Comments and Suggestions for Authors
The research manuscript explores the use of an intelligence-based method to select metabolites for distinguishing between open-angle glaucoma patients, type 2 diabetes patients, and healthy controls. The study uses metabolomics data from the aqueous humor and applies linear and non-linear discriminant analysis to identify key metabolites. The key findings include the selection of four significant metabolites—acetyl-ornithine (Ac-Orn), C3 acyl-carnitine (C3), diacyl C42:6 phosphatidylcholine (PC aa C42:6), and C3-DC (C4-OH) acyl-carnitine (C3-DC (C4-OH))—with a 95% discriminative success rate. The addition of a non-linear combination of these metabolites achieves 100% success. The study emphasizes the potential of this approach in developing a novel diagnostic tool and shifting focus from single-molecule biomarkers to combinations of metabolites.
-
Major Issues:
-
1. The manuscript lacks an in-depth discussion of the biological significance of the selected metabolites and their potential role in disease mechanisms. This information is crucial for readers to understand the relevance of the findings.
-
2. While the study reports 100% accuracy in classification, there is a lack of external validation or application of the proposed method to an independent dataset. The robustness and generalizability of the model should be assessed.
-
3. The introduction briefly touches upon the importance of biomarker discovery but lacks a comprehensive review of existing methods and their limitations. A more thorough discussion of related work would provide context for the proposed approach.
-
4. The manuscript mentions that the study has been evaluated by an ethics committee but lacks specific details on ethical approval, which is essential information for transparency and credibility.
-
5. The manuscript could benefit from a more detailed explanation of the statistical methods employed, especially regarding the rationale for selecting certain variables and the choice of non-linear combinations.
-
6. The scatterplots illustrating the relationships between variables (Figure 4) lack clear labels and explanations, making it challenging for readers to interpret the graphical data.
-
7. The conclusion section is concise and could be enhanced by summarizing the key implications of the findings and suggesting potential avenues for future research.
Minor Issues:
-
1. Some formatting issues, such as inconsistent font sizes and spacing, need attention. Additionally, citation details are marked as "To be added by editorial staff," which should be completed before publication.
-
2. The keywords section is cut off, and the missing keywords should be provided for indexing purposes.
-
3. The availability of data mentioned in the manuscript should be clearly specified, providing details on how readers can access the dataset.
-
4. Certain points, especially related to the methodology, are repeated in different sections. Streamlining the content for clarity is recommended.
-
5. Some sentences are complex and could be simplified for better readability. Clarity in presenting technical details is essential for a diverse readership.
-
6. The summary lacks explicit mention of the specific findings related to the discriminating metabolites and their combinations.
-
-
-
Author Response
The research manuscript explores the use of an intelligence-based method to select metabolites for distinguishing between open-angle glaucoma patients, type 2 diabetes patients, and healthy controls. The study uses metabolomics data from the aqueous humor and applies linear and non-linear discriminant analysis to identify key metabolites. The key findings include the selection of four significant metabolites—acetyl-ornithine (Ac-Orn), C3 acyl-carnitine (C3), diacyl C42:6 phosphatidylcholine (PC aa C42:6), and C3-DC (C4-OH) acyl-carnitine (C3-DC (C4-OH))—with a 95% discriminative success rate. The addition of a non-linear combination of these metabolites achieves 100% success. The study emphasizes the potential of this approach in developing a novel diagnostic tool and shifting focus from single-molecule biomarkers to combinations of metabolites.
Major Issues:
- The manuscript lacks an in-depth discussion of the biological significance of the selected metabolites and their potential role in disease mechanisms. This information is crucial for readers to understand the relevance of the findings.
Answer. We appreciate the comments that has been addressed in the revised version of the manuscript. We have modified some sentences as one of the messages we would like to convey is that not necessarily all selected metabolites are important for disease. Some may be important in terms of disease mechanisms; some may be important for diagnosis/prognosis and some for both. A typical example is, in blood, glucose versus glycosylated hemoglobin; one is important for both diagnosis and mechanism and the other is important for disease management, but not for mechanisms related to pathology. Moreover, in previous papers in which we used the same data, we were interested in looking for differentially concentrated metabolites and why they could give insight into disease mechanism. In the present study, we use the data to develop a novel methodology for disease diagnosis and management considering the perspective of personalized medicine.
- While the study reports 100% accuracy in classification, there is a lack of external validation or application of the proposed method to an independent dataset. The robustness and generalizability of the model should be assessed.
Answer. We understand the comment. Unfortunately, it is not possible to make an external validation at this point because the type of sample, aqueous humour, is not readily available. We think that the robustness comes from the fact that all the possibilities have been tested, even if it is in silico. It took an enormous amount of calculation time to try all the combinations and select the one that provided the fewest “errors.” This is what connects our method with AI because it will be a matter of selecting a few parameters (for example, from blood/serum tests) to at least i) make an accurate diagnosis, ii) stratify patients for a given disease, iii) optimize disease management and iv) allow a personalized form of therapeutic action. There is still work ahead that can bear fruit and it will be possible to carry out an unbiased evaluation that helps professionals and allows for truly personalized/precision medicine.
- The introduction briefly touches upon the importance of biomarker discovery but lacks a comprehensive review of existing methods and their limitations. A more thorough discussion of related work would provide context for the proposed approach.
Answer. Thanks for the suggestions; we have expanded the introduction to present the limitations of current methods.
- The manuscript mentions that the study has been evaluated by an ethics committee but lacks specific details on ethical approval, which is essential information for transparency and credibility.
Answer. We have included the requested information in the revised version of the paper; the committee, that is controlled at the regional government levels, ruled that the study did not require specific protocol to be approved. It should be noted that this study takes advantage of raw data from papers already published and in which this information was provided.
- The manuscript could benefit from a more detailed explanation of the statistical methods employed, especially regarding the rationale for selecting certain variables and the choice of non-linear combinations.
Answer. We have tested almost any possible combination in linear and non-linear fashion. Then we have tested every combination using the leave-one out strategy. We have tried to explain it better. In addition, and as per a similar suggestion and a request of the reviewer 2, we have included boxplots displaying the concentration of the four selected metabolites in the “first algorithm” (Figure 3 of the revised manuscript) and boxplots displaying the concentration of the six selected variables in the “second algorithm” (Figure 6 of the revised version). With these plots and the ad hoc explanations, the manuscript has substantially improved in quality, as we can visually explain why it is needed the discrimination power of a “combination of variables”, a single variable would not be sufficient. We have included both descriptions in the updated version. In addition, we have also included a correlation analysis of the selected metabolites.
- The scatterplots illustrating the relationships between variables (Figure 4) lack clear labels and explanations, making it challenging for readers to interpret the graphical data.
Answer. Thanks for spotting this. We have included the figures as supplementary material, while the underlying data have been included in two new tables. The information in table format is, in our opinion, much better to present and to explain. Thanks.
- The conclusion section is concise and could be enhanced by summarizing the key implications of the findings and suggesting potential avenues for future research.
Answer. We have totally rewritten the conclusion section including these suggestions that we appreciate.
Minor Issues:
- Some formatting issues, such as inconsistent font sizes and spacing, need attention. Additionally, citation details are marked as "To be added by editorial staff," which should be completed before publication.
Answer. We use Mendeley and the right format for the journal. Be assured that the journal will make all necessary modifications before the paper went public.
- The keywords section is cut off, and the missing keywords should be provided for indexing purposes.
Answer. Thanks for noticing. This has been amended in the revised version.
- The availability of data mentioned in the manuscript should be clearly specified, providing details on how readers can access the dataset.
Answer. This issue has been addressed in the revised version of the paper.
- Certain points, especially related to the methodology, are repeated in different sections. Streamlining the content for clarity is recommended.
Answer. This issue has been addressed in the revised version of the paper.
- Some sentences are complex and could be simplified for better readability. Clarity in presenting technical details is essential for a diverse readership.
Answer. This issue has been addressed in the revised version of the paper.
- The summary lacks explicit mention of the specific findings related to the discriminating metabolites and their combinations.
Answer. This issue has been addressed in the revised version of the paper
Modifications are in red lettering. Minor changes are not highlighted.
Round 2
Reviewer 3 Report
Comments and Suggestions for Authors
The authors have addressed my concerns.
Reviewer 4 Report
Comments and Suggestions for Authors
The authors' responses have been reviewed and found to be acceptable. The revised manuscript has significantly improved its merit.
Comments on the Quality of English LanguageMinor editing of English language required